# Sarcasm Detection over Social Media Platforms Using Hybrid Ensemble Model with Fuzzy Logic

**Dilip Kumar Sharma [1], Bhuvanesh Singh [2], Saurabh Agarwal [3], Nikhil Pachauri [4], Amel Ali Alhussan [5] and Hanaa A. Abdallah [6,\*]**

1 Department of Computer Engineering and Application, GLA University, Mathura 281406, India
2 Graduate Software Programs, University of St. Thomas, St. Paul, MN 55105, USA
3 Amity School of Engineering & Technology, Amity University Uttar Pradesh, Noida 201313, India
4 Department of Mechatronics, Manipal Institute of Technology, Manipal Academy of Higher Education, Manipal 576104, India
5 Department of Computer Sciences, College of Computer and Information Sciences, Princess Nourah bint Abdulrahman University, P.O. Box 84428, Riyadh 11671, Saudi Arabia
6 Department of Information Technology, College of Computer and Information Sciences, Princess Nourah Bint Abdulrahman University, Riyadh 84428, Saudi Arabia
\* Correspondence: haabdullah@pnu.edu.sa

**Abstract:** A figurative language expression known as sarcasm implies the complete contrast of what is being stated with what is meant, with the latter usually being rather or extremely offensive, meant to offend or humiliate someone. In routine conversations on social media websites, sarcasm is frequently utilized. Sentiment analysis procedures are prone to errors because sarcasm can change a statement's meaning. Analytic accuracy apprehension has increased as automatic social networking analysis tools have grown. According to preliminary studies, the accuracy of computerized sentiment analysis has been dramatically decreased by sarcastic remarks alone. Sarcastic expressions also affect automatic false news identification and cause false positives. Because sarcastic comments are inherently ambiguous, identifying sarcasm may be difficult. Different individual NLP strategies have been proposed in the past. However, each methodology has text contexts and vicinity restrictions. The methods are unable to manage various kinds of content. This study suggests a unique ensemble approach based on text embedding that includes fuzzy evolutionary logic at the top layer. This approach involves applying fuzzy logic to ensemble embeddings from the Word2Vec, GloVe, and BERT models before making the final classification. The three models' weights assigned to the probability are used to categorize objects using the fuzzy layer. The suggested model was validated on the following social media datasets: the Headlines dataset, the "Self-Annotated Reddit Corpus" (SARC), and the Twitter app dataset. Accuracies of 90.81%, 85.38%, and 86.80%, respectively, were achieved. The accuracy metrics were more accurate than those of earlier state-of-the-art models.

**Keywords:** BERT; fuzzy logic; GloVe; social media; sarcasm detection; Word2Vec

## 1. Introduction

The social media era is currently in effect. It has transformed communication around the world. Individuals may now exhibit themselves with the aid of social media with the tap of a finger. It is frequently used to express thoughts, feelings, and support on any subject or image posted via a social networking program. Daily sarcasm on social media apps such as Facebook and Twitter is familiar. A language device known as sarcasm is used to express contempt or negative emotions. It is a form of faux politeness that unintentionally works to make people angry. Sarcasm might appear to be unkindness with a thin veneer of deception. According to recent research, teasers frequently believe their comments are not as harmful as their victim perceives them to be [1]. However, in truth, they are more

damaging. Political parties and celebrities are frequently the targets of critical remarks and tags because these people are seen to be influential.

Sarcasm has a connection to psychiatric conditions, including despair and anxiety. People who were depressed or anxious during the pandemic utilized more sarcasm throughout their social media conversations, according to Rothermich [2]. In a face-to-face discussion, sarcasm may be quickly identified by paying attention to the speaker's gestures, tone, and facial expressions. However, because none of these characteristics are apparent in written communication, sarcasm can be difficult to spot. Sarcasm in pictures uploaded on social media platforms is even harder to spot because the context is either in the image itself or in the primary text/comment/headline. For false news detection, opinion mining, sentiment analysis, identifying cyberbullies, identifying online trolls, and other tasks such as these, sarcasm detection is essential. The capacity to detect sarcasm in social network apps, conversation boards, and e-comm sites is critical [3,4]. Sarcasm detection is currently a popular area of study. Sarcasm also impacts false news detection [5,6].

*1.1. Challenges in Sarcasm Detection*

Thanks to a vast amount of data, corporations have an excellent opportunity to learn further about people's perspectives and feelings, as well as other elements about them. However, there are also several challenges. For instance, sarcasm typically uses positive language, but because of the context, it conveys negative feelings. These minute issues have caused incorrect evaluation of product reviews in assessment analysis or incorrect categorization in false news identification. Many businesses and academics are now interested in obtaining accurate information from text data, including assertions made sarcastically, resulting from these issues. Many NLP algorithms are being offered, and although sarcasm detection is being trained for, these techniques also take context into account. The closeness of words helps us learn the contextual component. Different methods have various temporal proximities and are more closely tied to specific training contexts. Therefore, employing just one of them will not address all of the issues. Samples of sarcastic tweets are given in Figures 1–4. All the examples are taken from the Twitter platform.

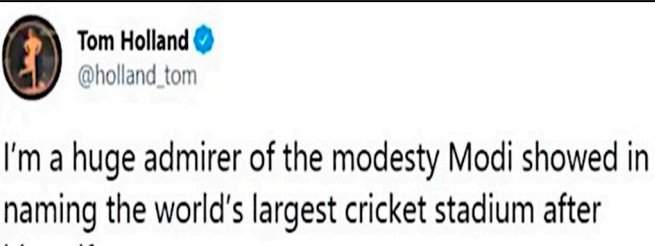

**Figure 1.** Tweet about a politician.

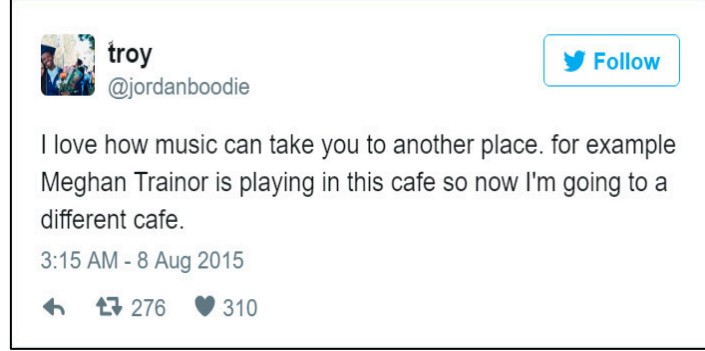

**Figure 2.** Tweet about a celebrity.

dentist: open up please
me: sometimes I get sad

**Figure 3.** Tweet with no hashtags.

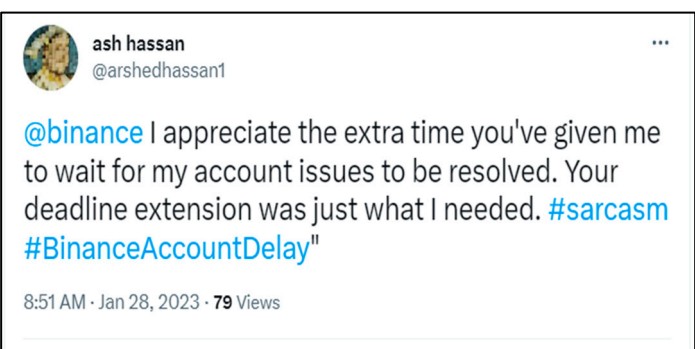

**Figure 4.** Tweet with positive hashtags.

Examples of sarcastic tweets against politicians and celebrities without the use of hashtags, emojis, or quotation marks are shown in Figures 1 and 2. These tweets use terms such as "big admirer", "love", and other positive adjectives, but the context is quite different. Sarcasm and favorable terms are used in Figures 3 and 4, but there are other linguistic elements such as exclamation marks, hashtags, and question marks. As a result, there are many different sorts of sarcastic remarks on social media, so each needs to be addressed for better identification. Due to the size restrictions on social media apps, people often use acronyms or slang in their remarks. It is challenging to understand these acronyms. In addition, it might not be easy to comprehend the context of the lingo and acronyms.

As sarcasm includes negative and positive terms, knowing the context is one of the critical difficulties in sarcasm identification. The second is coping with various content, such as hashtags, emojis, exclamation points, question marks, slang, and acronyms. Third, there is the need to find a way around the problem that one NLP approach cannot handle all the different kinds of material.

*1.2. Major Contributions*

The research community has undertaken several different attempts to address these issues, by relying either only on content-based characteristics or on context-based characteristics. To enhance the uncovering of sarcasm on social media sites, the problems given above must be resolved. A hybrid strategy must be suggested to address these various problems. The constraints of each prior study's use of a single content- or context-based method can be solved if we combine these strategies and address the issues. The authors of this study suggest a hybrid approach in response to this idea.

A hybrid ensemble model is proposed that merges training and classification from three context-based models: global vectors for word representation (GloVe), Word2Vec, and Bidirectional Encoder Representations from Transformers (BERT). It is to be noted that context is based on the words surrounding the center word in the training phase. BERT, a transformer, evaluates the context of the phrase rather than the words themselves and differs from word-based context models. The three models are trained, and the classification

probabilities from all three components are passed to the fuzzy layer, which has the final say in the classification. The information learned from the three models is weighted as high, medium, or low, centered on the score of embedding vectors provided by the respective model. The fuzzy logic module comprehends these weights and classifies whether a statement is sarcastic or not based on fuzzy rules. The intent is to use three different models, word-based and sentence-based, to cover most of the intricacies and let fuzzy logic balance out their limitations. On three publicly accessible social-media-focused datasets, accuracy scores of 90.81%, 85.38%, and 86.80% were attained. These accuracy levels surpass those of past frameworks put forward by scholars.

The following are the most important aspects of this research:

- The development of a credible and efficient hybrid ensemble solution to identify sarcastic texts on social media platforms.
- The use of hybrid models—word-based and sentence-based—in combination with the fuzzy logic module to compensate for their drawbacks is what makes this new approach novel.
- Twitter and Reddit are two examples of publicly accessible real-world social media datasets against which the model was evaluated.
- The model may be utilized on a variety of social media platforms and has a universal application.

The suggested framework has a wide range of practical and business applications. Businesses and academics may use the recommended method to accurately discern customers' true feelings and thoughts from evaluations of their products. This can assist them in removing caustic remarks that were mistakenly labeled good by earlier attempts. Political parties' IT departments can use this method to understand the genuine feelings of the public without being misunderstood since sarcasm is present. Political groups can create effective plans if there are lessons to be learned or new tactics to be used due to public feedback. Fact-checking companies who manually seek false news might use the model in their research to avoid having to look for sarcastic remarks. Currently, businesses and their workers are highly vocal in sharing their opinions on their firms on online forums. A business might take another look at these caustic remarks and provide suggestions regarding any poor choices or strategies made for personnel. Sarcastic remarks can also cause open surveys on any political, social, or cultural issue to produce false findings. Eliminating sarcastic remarks would enhance the interpretation of the survey's overall performance.

The remainder of this paper is structured as follows. Section 2 reviews significant studies on sarcasm that employ various methodologies. The proposed model is defined in detail in Section 3. The dataset, experimental discoveries, and evaluations are all included in Section 4. Section 5 concludes the paper and offers guidance for additional study.

## 2. Related Work

Despite sarcasm having long been studied in the social sciences, automated detection of sarcasm in text is a new area of study. Recently, the research community in the domain of natural language processing (NLP) and machine learning (ML) has been interested in automated sentiment classification [7]. An NLP-based technique uses linguistic corpora and language features to understand qualitative data. ML systems utilize unsupervised and supervised classification approaches based on tagged or unlabeled information to interpret sarcastic remarks.

An extensive dataset for sarcasm text detection was created by Khodak et al. [8]. Before comparing their findings with methods such as sentence vectorization, bag-of-words, and bag-of-bigrams, the authors performed hand annotation. They discovered that hand sarcasm detection outperformed other methods. Eke et al. [9] evaluated a range of earlier studies on sarcasm recognition. According to this review research, N-gram and part-of-speech tag (POS) methods were the most frequently used feature extraction algorithms. Binary interpretation and word frequencies were used for feature representation, nevertheless. The review also noted that the chi-squared test and information gain (IG) method were frequently employed for feature selection. In addition, the maximum entropy, naive Bayes, random forests (RF),

and support vector machine (SVM) classification techniques were used. A review of several "Customized Machine Learning Algorithms" (CMLA) and "Adapted Machine Learning Algorithms" (AMLA) utilized in sarcasm detection research was also published by Sarsam et al. in [10]. Their findings concurred with those of Eke et al. They found that CNN-SVM can perform better when both lexical and personal characteristics are used. The SVM performs better using lexical, frequency, pragmatic, and part-of-speech labeling.

Prior to this, models based on machine learning were created; these models primarily acquire language features and train these qualities over classifiers learned by machine learning. Machine learning has been used for content-based features by Keerthi Kumar and Harish [11]. Before submitting the data to the clustering method for various filters, the authors used feature selection approaches such as "Information Gain" (IG), "Mutual Information" (MI), and chi-square. An SVM was used to categorize data at the very end. Pawar and Bhingarkar [12] employed an ML classification model for sarcasm detection on a related subject. They collected data on recurring themes, punctuation, interjections, and emotions. Using the random forest and SVM techniques, these feature sets for classification were learned.

A different strategy was to go beyond individual words and understand the meaning of the sentences. "Long short-term memory" (LSTM), bidirectional LSTM, gated attention modules, or directed attention might be used primarily to perform this task. Sarcasm in Twitter posts was identified using neural network architecture by Ghosh and Veale [13]. They created an architecture using CNN and bidirectional LSTM. There were input data embeddings offered; the first were data from Twitter, and the second were author-related data. Word vectors were sent to bidirectional LSTM after being routed through CNN layers for feature learning. Dense layers received the bidirectional LSTM output vectors and used the SoftMax function to perform classification. Similarly, Ghosh, Fabbri, and Muresan [14] identified sarcastic comments using multiple LSTM and contextual data. Before judging whether or not anything was sarcastic, they studied the last tweets to grasp the perspective of the present statement. Liu et al. [15] used tweets to detect sarcasm and solely utilized content elements such as POS, punctuation, numerical data points, and emoticons. To identify sarcasm, Misra and Arora [16] employed a bidirectional LSTM, supplemented by an additional attention module. The module supplemented the LSTM by supplying the appropriate weights for the keywords under attention, while the bidirectional LSTM added context, taking into account the previous and following phrases. Additionally, they produced a new database of headlines from HuffPost and the website The Onion. In their unique strategy, Xiong et al. [17] combined self-matching words with a bidirectional LSTM model. When using self-matching phrases, standard information was ascertained by matching the words inside the phrases. A low-rank bilinear pooling technique was used to integrate composition data and inconsistencies to account for any information redundancy without impacting the classification results. Akula and Garibay [18] provided a multiheaded self-attention framework to classify sarcastic comments on various social media platforms. In order to calculate the distance correlation of words produced by the self-attention module, it employed gated recurrent units. Another multiheaded attention model using bidirectional LSTM was proposed by Kumar et al. [19]. The RoBERTa transformer, bidirectional LSTM, and a thinner BERT base were also utilized. Using the original method, they transferred the RoBERTa embeddings and bidirectional LSTM to the max pool layer. For sarcasm detection, Sundararajan et al. [20] applied a feature ensemble framework with a rule-based method. Another method of understanding the context is using a transformer-based approach. A contextual feature-based BERT model for identifying sarcastic tweets was put forward by Babanejad et al. in [21]. Potamias et al. suggested the RCNN-RoBERTa model, which is a transformer-based approach [22]. Parameswaran et al. [23] recommended combining an ML classifier and a DNN model to identify the object of sarcasm in text. They started by utilizing ML to classify sarcastic sentences and determine if they included a target (using LSTM). A DNN algorithm using aspect-based sentiment was then used to retrieve the target. According to Du et al. [24], examining context is essential for sarcasm

identification. Context should contain the attitudes expressed in comments in response to the specific language material and the participant's expressive habits. They suggested two-stream CNN, which assesses the emotional component and semantics of the target language text. SenticNet augments the "long short-term memory" (LSTM) concept with common sense. The user's attention mechanism considers the user's expressive behaviors. The authors of [25] used BERT, LSTM, and an autoencoder-based hybrid model for sarcasm detection. Pandey and Singh [26] also suggested a BERT-LSTM-based algorithm for sarcasm detection for code-mixed social network tweets. They used a pretrained BERT model and passed the output vectors through the LSTM layer. A code-mixed dataset was employed for the validation of the model. Similarly, Savini and Caragea [27] employed BERT. They employed a trained BERT model but fine-tuned the architecture by adding intermediate and target tasks before classification.

Other studies used additional modalities such as user behavior, hashtags, personality factors, and emotions to identify sarcasm. Hazarika et al. [28] presented context- and content-based embedding approaches for sarcasm classification on social media. The model used user features, such as users' personalities and stylometric characteristics. An approach using character-level feature representations of words was presented by Illic et al. [29]. The foundation was "Embeddings from Language Models" (ELMo). Malave and Dhage [30] published an approach for recording user behavioral patterns, personal traits, and contextual data for sarcasm identification. The hashtag model was proposed by Sykora, Elayan, and Jackson [31] and is based on social network apps. It examined the hashtags' characteristics and recognized sarcastic phrases. Yao et al. [32] used a highly original method. They used text and photos from Twitter that were based on four different content contexts to identify sarcasm. They employed picture captions, text over images, tweets with images, and tweets without images. A multichannel interaction approach centered on gated and directed attention modules was used to teach these multimodalities. A unique emotion-based paradigm for sarcasm classification was proposed by Agrawal, An, and Papagelis [33]. On the other hand, most modern models consider emotional traits (joy, surprise, or sadness), ignoring the sequence of information retained between affective states. The authors used a sequence classification approach to the problem of sarcasm detection, taking advantage of the variations in diverse emotions that naturally occur throughout a text to observe the effects of alterations in emotional states.

Studies have also been carried out in languages other than English. Sarcasm among native and non-native English speakers was examined by Techentin et al. [34]. They discovered that several experience-related characteristics influence non-native speakers' capacity to recognize and employ ironic cues. In 2020, Farha and Magdy [35] researched Arabic sarcasm recognition. A model for sarcasm recognition across Hindi-English tweets was developed and used by Swami et al. [36] in a comparable pattern. The authors of [37–39] worked on tweets and YouTube comments, and much work still needed to be performed for sarcasm detection in the data. Kamal and Abulaish [40] suggested a model for self-deprecating sarcasm (SDS). SDS is sarcasm directed at the self. The authors proposed a model with a convolutional layer along with a bidirectional gated recurrent unit. Further output vectors from the bidirectional gated recurrent unit were passed through two attention layers. Sigmoid was used to classify the final layer. They validated their model against seven Twitter datasets. Elkamchouchi et al. [41] employed a hybrid of the evolutionary model known as the hosted cuckoo optimization algorithm and a machine learning model with stacked autoencoders for sarcasm detection. They validated against the Kaggle dataset. The architecture was stepwise-based. After initial preprocessing, TF-IDF was used to generate word embeddings. The embeddings were then passed to a hosted cuckoo optimization algorithm for parameter tuning. The classification was performed using stacked autoencoders.

The problem with the previously cited study is that it all focuses on a single strategy. There are restrictions when using context-based or unique content techniques. Given that context and content vary depending on the various networks, there might not be a universal

answer. In particular studies, several feature ensemble frameworks have been proposed; nevertheless, the suggested models correctly combine all modalities from all features. Using features other than content and context is not practical as a fix. User behaviors and personality traits are particular to each individual and cannot be standardized to create a scalable social platform solution.

The authors of this study aim to solve these issues by creating a hybrid model that considers both word and sentence contexts (using a BERT transformer). The novel feature of the suggested approach is the fuzzy logic controller, which regulates the limits in various ways based on the rules. The solution is universal because many tactics are employed, and their teachings are given the proper weight.

## 3. Proposed Framework

The suggested framework is a hybrid ensemble method that uses content-based approaches to offer the sarcasm detection method the most specific solution. To determine whether a tweet or text is ironic, the model uses GloVe, Word2Vec, and BERT-base. The fuzzy logic layer evaluates the classification likelihood of all three procedures using the outlined section from the three previously described techniques. The tweets or text are ultimately classified as sarcasm depending on rule-based fuzzy logic.

The flowchart of the proposed model is shown in Figure 5. The preprocessing layer is the first layer to receive text input. After the readers have gone through preprocessing, the exact text is transmitted to three different components, namely Word2Vec, GloVe, and BERT. Each component creates a vector space of word/sentence embeddings, but the results vary since each employs a distinct embedding creation method. Dense layers receive embeddings to learn feature vectors. The dense layer output from each component is transferred to the SoftMax layer. Each SoftMax layer passes its classification probability to the fuzzy logic component for final classification. The rule-based fuzzy logic layer module gives the three categorization probabilities weights. Based on the prescribed fuzzy rules, it finally marks them as sarcastic or genuine.

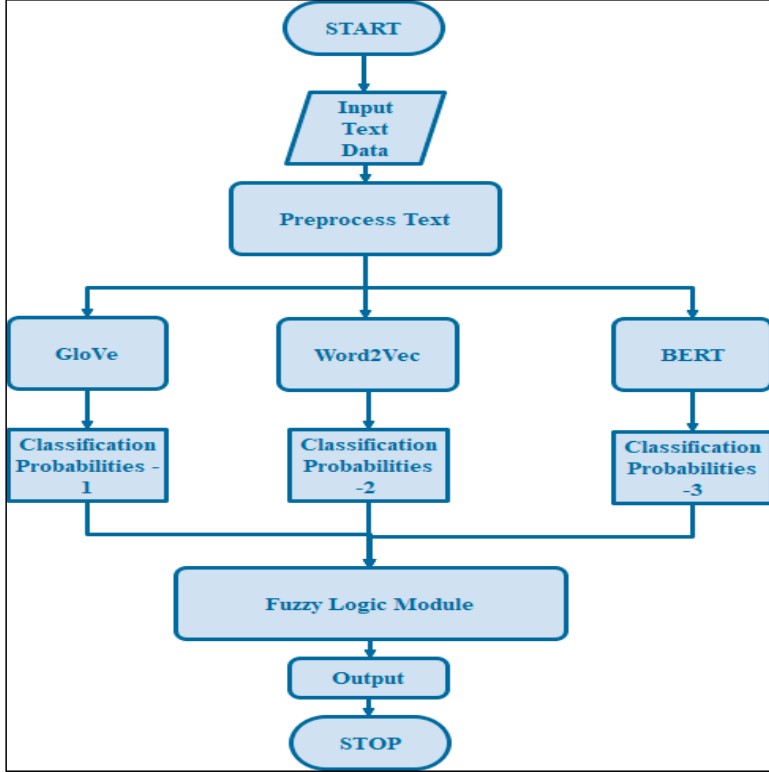

**Figure 5.** Flowchart of proposed framework.

A thorough representation of the suggested model is shown in Figure 6. We can observe how the input text is routed via several levels before being sent to their dense layers for learning. The probabilities of all three are sent to the third and final fuzzy layer. The word-embedding methods GloVe and Word2Vec convert words into vectors. The internal addition and subtraction of the words may be performed as they are vectors. BERT, on the other hand, creates sentence embeddings. After considering the entire text, BERT produces a vector. It also considers the preceding and succeeding phrases to provide context-driven sentence embeddings.

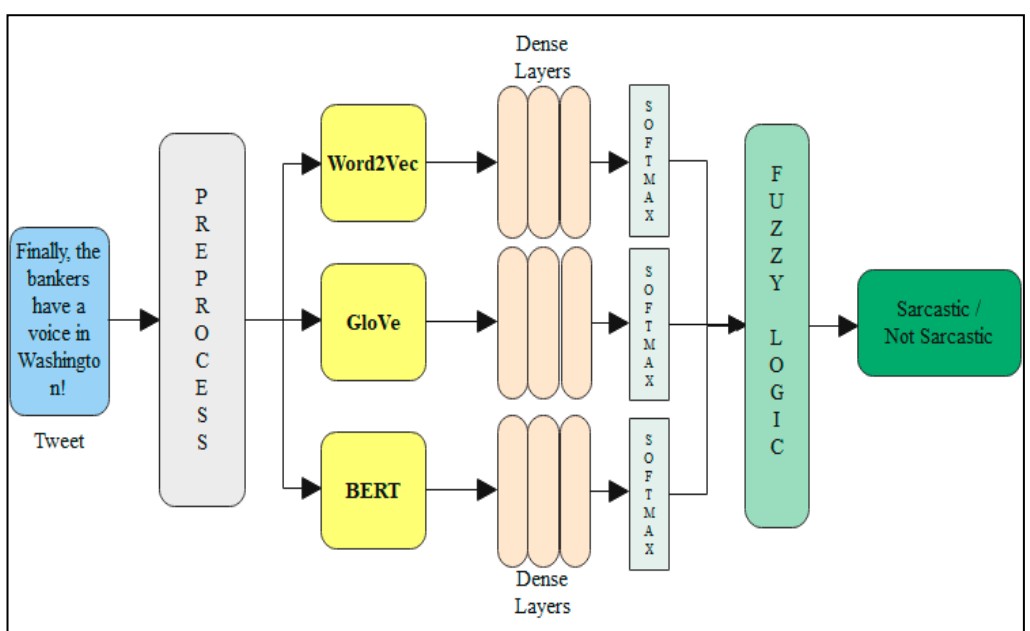

**Figure 6.** Architecture of proposed model.

### 3.1. Word Vectorization

Word vectorization involves transforming words into numerical vectors, or embeddings, which capture their meaning and relationships with other words. The main objective of word embedding is to maintain the contextual similarity of the corpus while downsizing the high-dimensional feature space into low-dimensional feature vectors. Word embeddings are a crucial part of several NLP strategies. They show how a computer can interpret spoken words. They can be considered representations of vectorized text. Word vectorization methods include techniques such as GloVe and Word2Vec.

#### 3.1.1. Global Vectors for Word Representation (GloVe)

GloVe is a Stanford-developed unsupervised learning technique that creates word vector representations. GloVe is a word-embedding technique that uses a co-occurrence matrix to compute word similarities in a corpus. It trains a regression model to learn word vectors that capture the relationships between words based on how often they appear together in the corpus. These word vectors are low-dimensional, dense representations that capture words' meanings and semantic relationships. GloVe works on two major components: nearest neighbors and linear substructures.

The cosine similarity (Euclidean distance) between two words is used to measure the neighborhood distance. The cosine similarity determines how similar two words are linguistically or semantically. The similarity metrics created in this neighborhood evaluation consist of a single scalar that gauges the relationship between two words. Since two sentences usually have more intricate connections than a single number can describe, this simplicity may need to be revised. The solution to this issue is using linear substructures. A model must link more than one number with the word pair to statistically capture the information needed to distinguish between two identical words. The two-word

vector difference is a clear and straightforward choice for inclusion in a more extensive collection of discriminative numbers. The central notion behind the model is centered on the simple insight that the ratio of word-to-word co-occurrence probability might carry some information. For example, one could consider the probability of the target terms ice and steam co-occurring with various probing words from the lexicon. Figure 7 shows the probability ratio table for the words ice, steam, and gas.

| Probability and Ratio | k = solid | k = water | k = gas | k = fashion |
|---|---|---|---|---|
| P(k\|ice) | $1.9 \times 10^{-4}$ | $6.6 \times 10^{-5}$ | $3.0 \times 10^{-3}$ | $1.7 \times 10^{-5}$ |
| P(k\|steam) | $2.2 \times 10^{-5}$ | $7.8 \times 10^{-4}$ | $2.2 \times 10^{-3}$ | $1.8 \times 10^{-5}$ |
| P(k\|ice) / P(k\|steam) | 8.9 | $8.5 \times 10^{-2}$ | 1.36 | 0.96 |

**Figure 7.** The table of probability ratios of GloVe working for a corpus of words [42].

The aim of GloVe is to capture sublinear connections in the vector space by forcing the word vectors. It performs better than Word2Vec in word analogy tasks as a consequence. GloVe increases the usefulness of word vectors by focusing on word-pair-to-word-pair connections rather than word-to-word interactions. Word co-occurrence matrices, which require a large amount of memory, are used in the training of the GloVe model. It takes a long time to reconstruct the matrix to change the hyperparameters connected to the co-occurrence matrices.

### 3.1.2. Word2Vec

For word representations in vector space, the Word2Vec model was developed in 2013 by Tomas Mikolov and numerous Google research teams. While GloVe uses a linear regression model, Word2Vec uses shallow neural networks to learn the continuous vector representations of words from large text corpora. It trains the model on a prediction task, such as predicting a target word given its surrounding context words (CBOW) or predicting the context words given a target word (Skip-Gram). The learned word vectors capture the semantic and syntactic relationships between words, allowing for their use in various NLP tasks. The resulting embeddings are compact, dense representations that can be used for further processing, such as clustering, classification, or semantic arithmetic. Word2Vec is a technique for generating word embeddings that may be applied to various applications, such as sentiment analysis, recommendation systems, and text similarity.

The power of Word2Vec comes from its capacity to aggregate vectors of similar words. When given a large enough dataset, Word2Vec can infer a word's meaning based on how frequently it appears in the text. These computations determine the relationships between words and other terms in the corpus. The probability of the context word concerning the center word is calculated as per Equation 1. For example, all words in the corpus could be denoted by two vector sets: Uw and Vw. Uw is the vector representing the context word, and Vw represents the center word. With these vectors, the probability equation on the central word p and context word c can be denoted with the equation below.

$$P(O = o \mid C = c) = \exp\left(U_o^T V_c\right) / \sum_{w \in Vocab} \exp\left(U_w^T V_c\right) \tag{1}$$

The success of Word2Vec may be attributed to two significant architectural designs: Skip-Gram and the "Continuous Bag of Words" (CBOW). The Skip-Gram model uses various words to predict the missing one given an input word, whereas the CBOW model uses a single word to predict the missing one. Figure 8 depicts how the two architectures work. Word2Vec implementation requires implementing Word2Vec using any of the architectures. The advantages of using Word2Vec are that data may be input into the model in real time and require little preparation. Thus, memory is not a concern. Word vectors may infer associations such as "king: man as queen: woman" because the mapping between the target item and its context word embeds the sublinear relationship into the vector space of

words. The "Skip-Gram model works well with a small amount of training data and can better catch unusual words or phrases", according to the original developers of Word2Vec. The CBOW model, in contrast, appears to learn faster than Skip-Gram and can represent more common words more accurately, resulting in greater accuracy for frequent keywords. We have used the CBOW method in our proposed framework. Sarcastic comments are common, and these common words are frequently used.

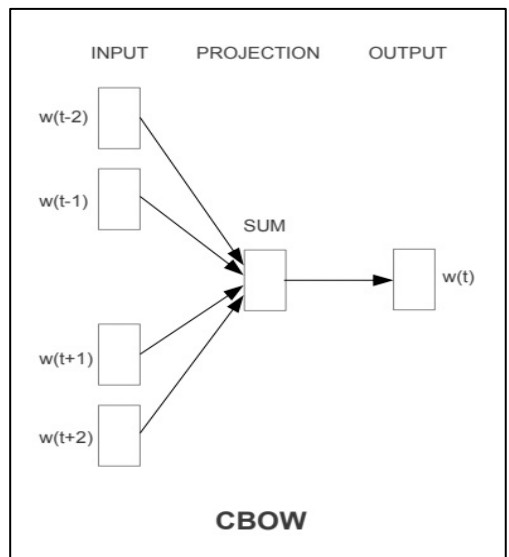 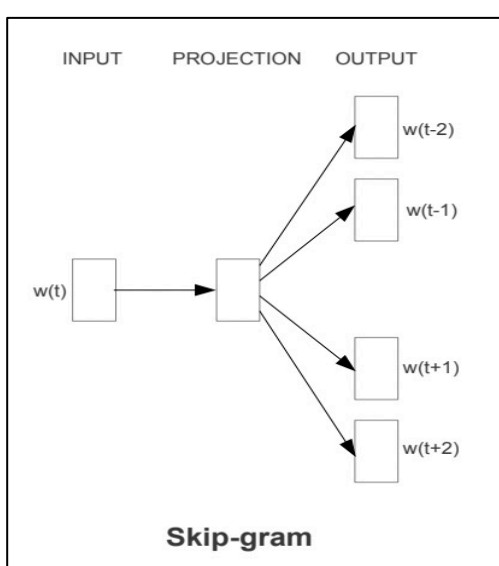

**Figure 8.** Two architectures, CBOW and Skip-Gram, are employed by Word2Vec [43].

*3.2. Sentence Embedding*

Sentence embedding is the process of converting a sentence into a numerical vector representation, also known as sentence vector. It is used to capture the semantic meaning and relationships between sentences in natural language processing tasks. Sentence embeddings can be generated by combining word embeddings with techniques such as averaging and concatenation, or more advanced models such as transformer-based networks. Sentence embeddings can be used in various NLP tasks such as text classification, text similarity, and machine translation. BERT is one such transformer-based method.

BERT

Word2Vec and GloVe are word-embedding architectures. They have problems with tasks such as learning the representation of terms that are not in their vocabulary and learning how to separate some word pairs that are opposed. For example, in the vector space, the words "good" and "bad" are generally extremely close to each other, which may restrict the effectiveness of word vectors in NLP applications such as sentiment analysis.

BERT is a self-supervised pretraining method that learns to anticipate intentionally disguised (masked) text sections. It was created and demonstrated by a Google Research team. Transformer architecture serves as the foundation for BERT. BERT is defined as follows by the research team for BERT-base: "BERT stands for Bidirectional Encoder Representations from Transformers". It was created to pretrain deep bidirectional interpretations from unlabeled text by simultaneously conditioning the left and right contexts. However, cutting-edge models for diverse NLP applications may be produced by fine-tuning the pretrained BERT model with just one additional output layer. GloVe and Word2Vec were unidirectional language models used for natural language processing in the past. It is taken into account that they could only understand the context of the object word in one way by sliding a window of "n" words (to the right or left of the object word). The BERT model may function in both ways. The BERT transformers may move sentences to the left and right to comprehend the target word's context fully.

The BERT-base architecture has twelve layers of encoders. The encoder approach is crucial since BERT aims to develop a language model. A 768-dimension embedding is a result. Its two main parts are Next Sentence Prediction (NPS) and masked language modelling (MLM). Thanks to masked LM, models that were previously difficult to train in may now be trained in both directions (MLM). This trait enables the model to deduce the context of a word from its surroundings (left and right of the word). NPS is used throughout the training phase. The training strategy involves feeding the model pairs of sentences in order to learn and predict whether the second sentence in a pair will occur after another in the original text. Half of the inputs during training are pairs of sentences, where the second sentence follows the first in the original text. The remaining 50% of the sentences are selected randomly from the corpus. From the randomly chosen statements, the first one will be made. It should be noted that BERT-base is a pretrained model that was developed using a Wikipedia corpus of 2500 million words. Figure 9 depicts the BERT-base architectural diagram.

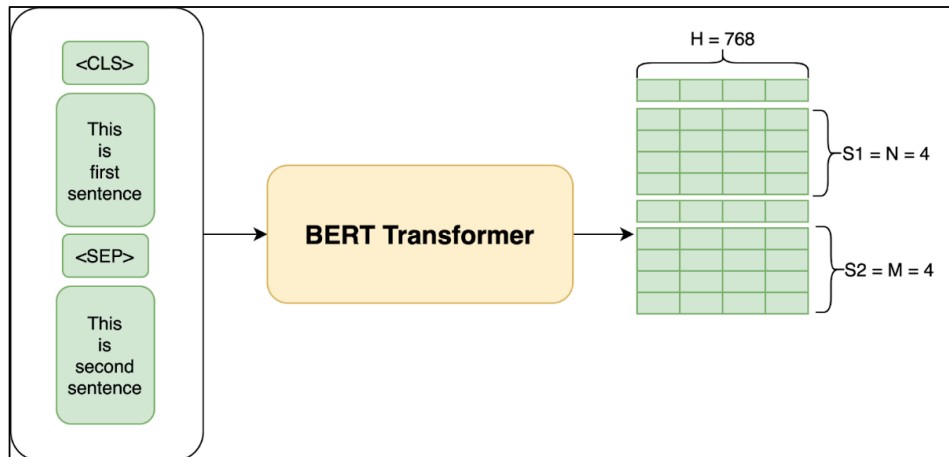

**Figure 9.** The architecture of BERT-base [44].

### 3.3. Fuzzy Logic

The following definition was provided by Charles Elkan, an assistant professor of computer science and engineering at the University of California, San Diego: "Fuzzy logic is a generalization of standard logic, in which a concept can possess a degree of truth denoted by 'μ' anywhere between 0.0 and 1.0. Standard logic applies only to completely true concepts (having a degree of truth of 1.0) or false (having a degree of truth of 0.0). Fuzzy logic is supposed to reason inherently vague concepts, such as 'tallness'. For example, we might say that 'President Clinton is tall', with a degree of truth of 0.9".

Rule-based fuzzy logic is a decision-making system that uses a set of if–then rules to make decisions based on uncertain or vague input data. It maps the input data to several fuzzy sets, representing degrees of membership for each input in a particular category. The system then applies the rules to determine the degree of membership for the output. The final output is obtained by aggregating the results of the individual rules and defuzzifying the result to obtain a crisp value. The link between the set's elements and their degree of truth is known as the membership function of the set. Figure 10 shows how membership functions may be applied to temperature.

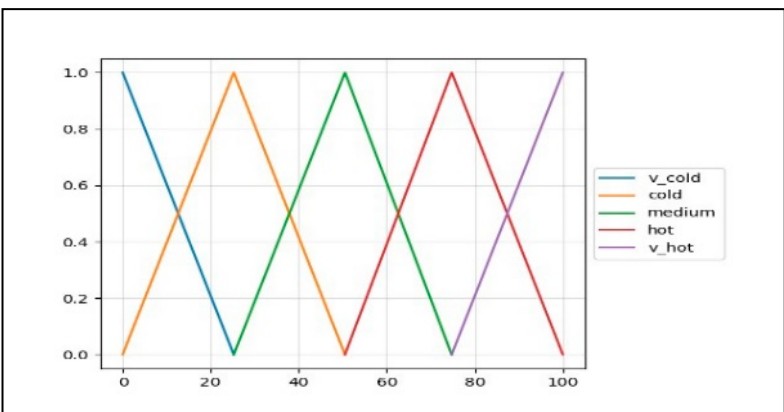

**Figure 10.** Membership function example for temperature [45].

A fuzzy system is a collection of fuzzy expertise and knowledge that can reason about data in general ways rather than using strict Boolean logic. Expert knowledge comprises a set of fuzzy membership functions and fuzzy rules.

The authors of this study applied fuzzy logic for the final classification in the current research paper. The output classification probabilities from three components, Word2Vec, GloVe, and BERT, are passed through the fuzzy logic module. The membership function is based on "high", "medium", and "low" values. Any classification probability above 0.75 is marked as high. Probability values between 0.40 and 0.75 are medium, and those below 0.4 are low. The fuzzy rules are dictated based on the high, medium, and low labels—these fuzzy rules balance out the pros and cons of the technique used at the individual level. Table 1 describes the working of the fuzzy logic controller. As the probability output from SoftMax includes low, medium, and high values, these values are passed to the fuzzy logic controller, where fuzzy logic rules are applied, and based on probability combinations, the final combination is predicted.

**Table 1.** Fuzzy logic controller rules.

| Fuzzy Logic Controller | Domain Knowledge | Examples of Fuzzy Rules |
| --- | --- | --- |
| $FLC_{Prob}$ | If a SoftMax probability value from Word2Vec is low, there is a low probability value from GloVe, and there is a low BERT value, then the final weighting unit from the fuzzy logic controller is likely to be low. Suppose a SoftMax probability value from Word2Vec is high. In that case, with a high probability value from GloVe, and a high BERT value, then the final weighting unit from the fuzzy logic controller is likely to be high. | If ($Word2Vec_{prob}$ is low) and ($GloVe_{Prob}$ is low), and ($BERT_{Prob}$ is low), Then ($FuzzyLogicController_{Prob}$ is low). If ($Word2Vec_{prob}$ is medium) and ($GloVe_{Prob}$ is medium) and ($BERT_{Prob}$ is medium), Then ($FuzzyLogicController_{Prob}$ is high). If ($Word2Vec_{prob}$ is high) and ($GloVe_{Prob}$ is high) and ($BERT_{Prob}$ is high,), Then ($FuzzyLogicController_{Prob}$ is high). |

### 3.4. Execution of Framework

After preprocessing, all the comments, tweets, and headlines are passed to each component: Word2Vec, GloVe, and BERT. For the provided text, each component builds an embedding. Because they construct embeddings using various methods, the results (embeddings) differ. The embeddings are given to the relevant dense layers for feature vector learning. The dense layer's output is promoted to the SoftMax layer for each component. The SoftMax layer outputs the probability of the tweet or sentence being fake or real. In this case, the probability values from each component are Word2Vecprob, GloVeProb, and BERTProb. The classification probabilities (Word2Vecprob, GloVeProb, and BERTProb) from each SoftMax layer are transferred to the fuzzy logic component for final classification. The fuzzy logic layer module weighs the three categorization probabilities using a rule-based approach.

Examples of the rules are provided in Table 1. The fuzzy logic controller determines whether the tweet or sentence is sarcastic based on the offered fuzzy principles.

*3.5. Challenges of Hybrid Models*

Hybrid models using ensemble techniques have specific challenges. First, they need help interpreting. There are multiple techniques, and results are throughput as a black box. Second, there are various techniques, so it takes a large amount of work to train and deploy these models. Third, the fine-tuning and customization of the framework is a complex task. The proposed framework overcomes these challenges. The authors of this study have utilized a rule-based fuzzy logic controller. This employment of fuzzy logic resolves two significant issues. First, as the rules can be defined and fine-tuned, the overall framework results can be fine-tuned and customized. For example, we can update the fuzzy rules if we observe greater accuracy and performance by fine-tuning high, medium, and low threshold limits. Second, as the final classification authority lies with the fuzzy logic controller, we know the interpretations and why this final classification has been performed. This makes the framework explainable. The authors utilized pretrained NLP models; GloVe, Word2Vec, and BERT-base are all pretrained on diverse content types. The embeddings from these pretrained models are passed through dense layers. Therefore, the pretrained models and dense layers can be customized with corresponding hyperparameters. Deployment of the pretrained models is also less complex and less computation-intensive than deploying a fully customized NLP model. Fuzzy logic is rule-based, making the entire proposed architecture lightweight, flexible, and robust.

**4. Experimental Results**

We carried out a comprehensive experiment on publicly accessible social media datasets—SARC [8], Twitter datasets [13], and Headlines datasets—to ensure the model's efficacy [16]. The model was created on a computer using the Google TensorFlow framework and the Keras library. The activation function "ReLU" and the optimizer "Adam" in the dense layers have a learning rate of 3e-4 each. Since this is a binary classification problem, binary cross-entropy was employed as the loss function. For each component, these hyperparameters remained constant across all dense layers. The dataset was 20% for testing and 80% for training. The best results were documented once the highest level of precision had been attained. The network was terminated using accuracy metrics.

*4.1. Datasets*

4.1.1. Twitter Dataset

For this dataset, tweets were compiled using Twitter online sarcasm. The dataset included tweets, tweet responses, and the tweeter's emotion at the moment of the tweet. Tweets and retweets from users served as the content, while tweets' responses served as the context. Out of 1956 tweets, 1061 were accepted as not sarcastic, and 895 were recognized as sarcastic by their writers.

4.1.2. SARC Dataset

Reddit forum comments are included in the self-annotated Reddit corpus collection known as SARC 2.0. The s token, used by users to signify sarcasm, erases their sarcastic comments. Our research only used the original remarks; parent or child comments were not included. The primary and politics-related dataset versions were used in our experiments; the latter contains comments from the political subreddit.

4.1.3. Headlines Dataset

Two news websites, The Onion and HuffPost, provided the news headlines for this dataset. HuffPost features accurate news headlines, while The Onion features satirical takes on current events. The news pieces served as context, and headlines served as content. There were 26,709 headlines in all. Out of these, 14,984 were not sarcastic, whereas 11,725 were.

### 4.2. Metric

To assess the performance of our model, we computed accuracy, precision, recall, and F1 score. The confusion matrix is depicted in Figure 11. These are the most often-used performance metrics for classification issues. The output from our models on the three publicly available datasets is shown in Table 2. The model's accuracy scores for the Headlines, SARC, and Twitter datasets were 90.81%, 85.38%, and 86.80%, respectively. The performance metric outperformed other earlier studies on sarcasm recognition. Tables 3–5 compare the findings of the developed framework with those of earlier research models on the SARC, Headlines, and Twitter datasets.

$$Precision = TP \div (TP + FP) \qquad (2)$$

$$Recall = TP \div (TP + FN) \qquad (3)$$

$$Accuracy = (TP + TN) \div (TP + TN + FP + FN) \qquad (4)$$

$$F1 - Score = 2 \times (Precision \times Recall) \div (Precision + Recall) \qquad (5)$$

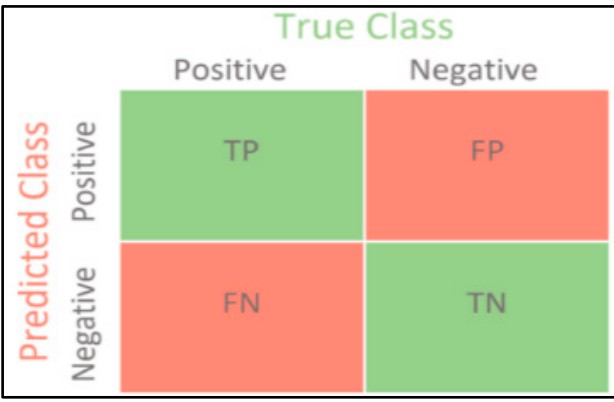

**Figure 11.** Confusion matrix.

**Table 2.** Performance metric of the proposed model.

| Proposed Model | Accuracy | Precision | Recall | F1 |
|---|---|---|---|---|
| SARC [8] | 85.38% | 0.8517 | 0.8558 | 0.8537 |
| Twitter [13] | 86.80% | 0.848 | 0.8833 | 0.8653 |
| Headlines [16] | 90.81% | 0.92 | 0.9115 | 0.9157 |

**Table 3.** Performance comparison of the SARC dataset.

| Models | Accuracy | Precision | Recall | F1 |
|---|---|---|---|---|
| SARC [8] | 77.00 | - | - | - |
| Multihead Attention [18] | 81.00 | - | - | - |
| RCNN-RoBERTa [22] | 79.00 | 0.78 | 0.78 | 0.78 |
| BERT+EmoNetSent [27] | - | - | - | 0.775 |
| CASCADE [28] | 74.00 | - | - | 0.75 |
| ELMo-BiLSTM [29] | 79.00 | - | - | - |
| Proposed | 85.38 | 0.85 | 0.85 | 0.85 |

**Table 4.** Performance comparison of the Twitter dataset.

| Models | Accuracy | Precision | Recall | F1 |
|---|---|---|---|---|
| Sarcasm Magnet [13] | 72.5 | 0.73 | 0.71 | 0.72 |
| Sentence-Level Attention [14] | 74.9 | 0.74 | 0.75 | 0.74 |
| A2 TextNet [15] | 80.1 | 0.83 | 0.80 | 0.80 |
| Self-Matching Network [17] | 74.4 | 0.76 | 0.72 | 0.74 |
| Multihead Attention [18] | 81.2 | 0.80 | 0.81 | 0.81 |
| Proposed | 86.8 | 0.84 | 0.88 | 0.86 |

**Table 5.** Performance comparison of the Headlines dataset.

| Models | Accuracy | Precision | Recall | F1 |
|---|---|---|---|---|
| A2 TextNet [15] | 86.20 | 0.86 | 0.86 | 0.86 |
| Hybrid [16] | 89.70 | - | - | - |
| Multihead Attention [18] | 91.60 | 0.91 | 0.91 | 0.91 |
| Proposed | 90.81 | 0.92 | 0.91 | 0.91 |

*4.3. Results Analysis*

This section discusses the analysis of the results obtained from various datasets using the proposed model. First, the various metrics collected and their comparisons with previous research are provided, and then the analysis of each dataset is explained.

4.3.1. Analysis of SARC Dataset

The SARC dataset, the largest dataset, was collected from the Reddit website. Table 3 demonstrates that preceding researchers mostly used attention and LSTM/bidirectional LSTM in their experiments [8,22,29]. Figures 12 and 13 illustrate the confusion matrix and results in the comparison table for accuracy, respectively. The BERT-base developers have shown that the BERT-base employed in our proposed model may be enhanced to LSTM and BiLSTM. This tendency may be attributed to the fact that discussion forum comments, such as those from Reddit, differ significantly from those from other social sites used for our intermediate tasks in terms of substance, expressiveness, and subject. For instance, SARC comments might range from three to four words to hundreds of words. Second, Reddit includes comments from both the main site and subreddits. It is crucial to comprehend the context of preceding sentences as a result. Since LSTM takes words within a sentence and does not take the entire phrase, LSTM/BiLSTM models may lose context and tend to have an overfitting effect. The bidirectional BERT-base is a transformer with a context-sensitive encoder stack that takes previous and following sentences. An enormous corpus with diverse domains is used to train the BERT-base. Consequently, our proposed model is more accurate than past models that employed LSTM due to the inclusion of the BERT-base.

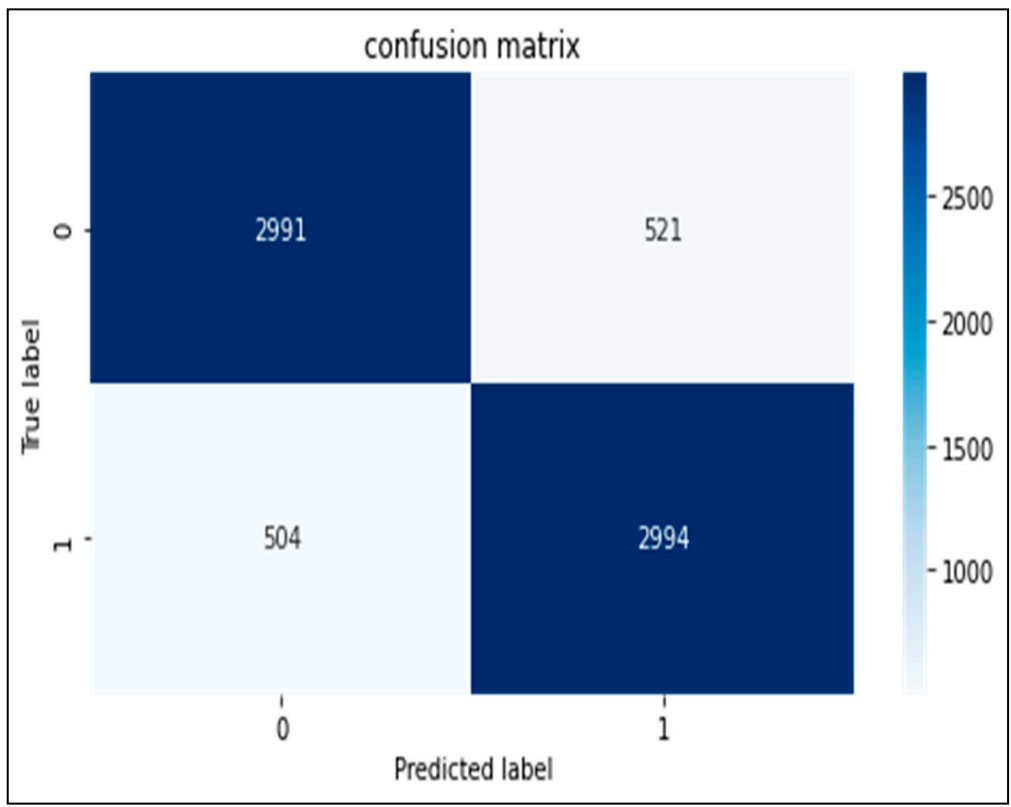

**Figure 12.** Confusion matrix results: SARC dataset.

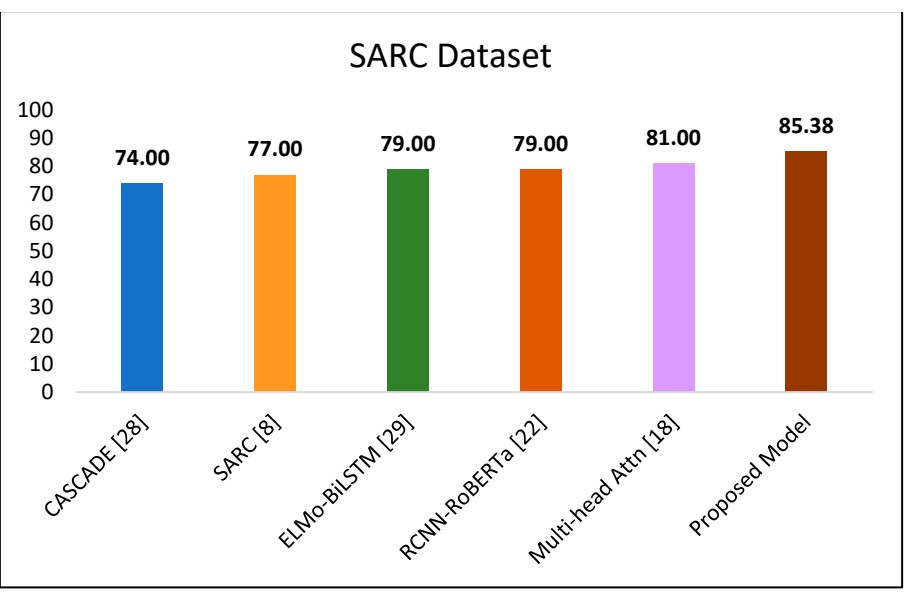

**Figure 13.** Accuracy evaluation: SARC dataset [8,18,22,28,29].

### 4.3.2. Analysis of Twitter Dataset

Most prior research has used an attention mechanism on Twitter datasets [14,17,18]. The confusion matrix and results in the comparison table for accuracy are shown in Figures 14 and 15, respectively. Table 4 displays a thorough comparison. The attention mechanism might help people focus on a single word or sentence. Twitter comments are frequently brief and use slang and acronyms. Because every post is distinct, a clear emphasis is impossible to achieve. The usage of slang and abbreviations can cause issues with the attention mechanism. Our model uses GloVe

and Word2Vec, which utilize words and work better on short sentences, providing better accuracy for the Twitter dataset.

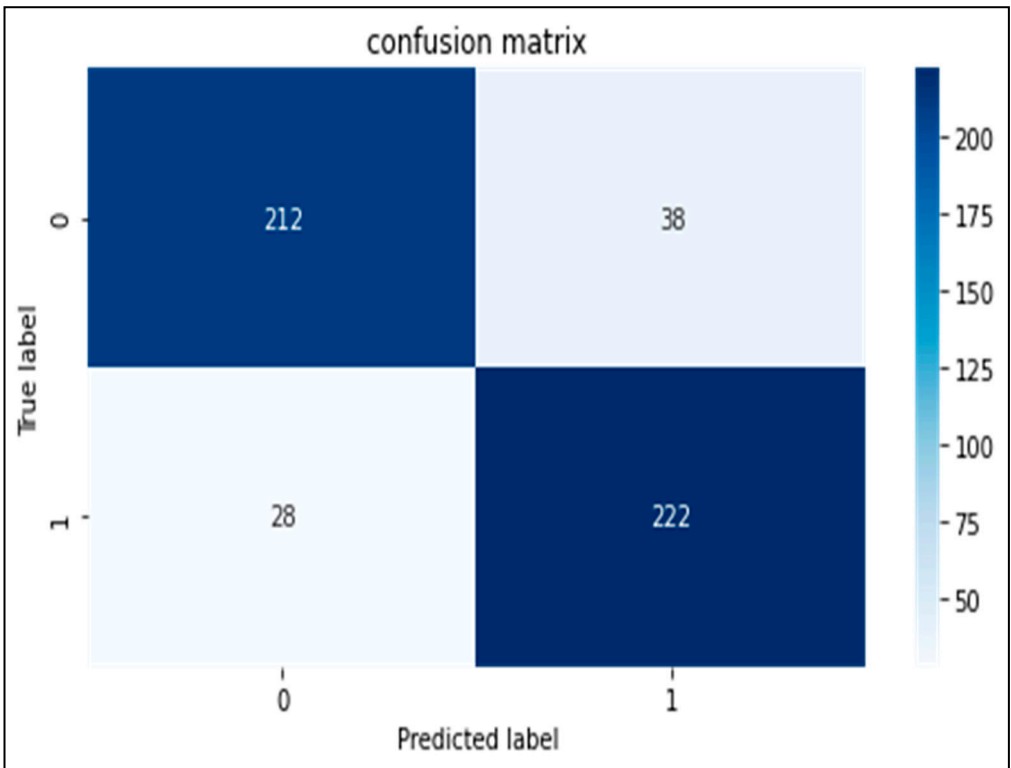

**Figure 14.** Confusion matrix results: Twitter dataset.

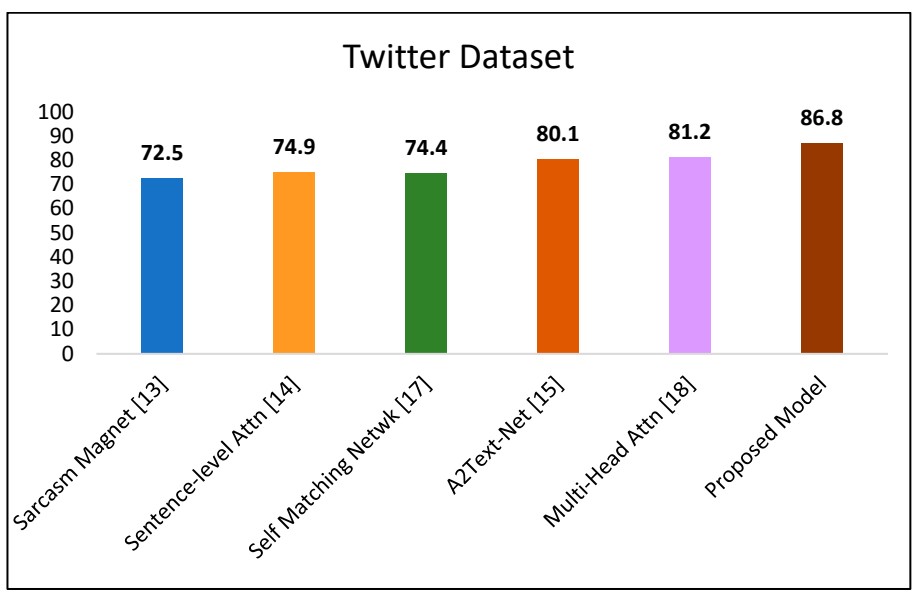

**Figure 15.** Accuracy evaluation: Twitter dataset [13–15,17,18].

### 4.3.3. Analysis of Headlines Dataset

Our proposed model's accuracy is more or less similar to that of previous models [16,18] on the Headlines dataset. However, there is a slight difference, with Akula and Garibay [18] having the highest accuracy. The comprehensive comparative findings are shown in Table 5. Figures 16 and 17 depict the confusion matrix and results in the comparison table for accuracy, respectively. This is due to the headlines' variety. News headlines do not have any previous or following sentences to allow the context to be understood

well. Every news headline is unique, and headlines are occasionally ironic. Given that there is no association between the previous and following sentences, sentence embedding and BERT may not perform similarly on the Headlines datasets. In this case, GloVe and Word2Vec perform better at detecting the usage pattern of specific terms in the Headlines dataset. Consequently, the proposed framework's accuracy is equivalent to that of past study frameworks.

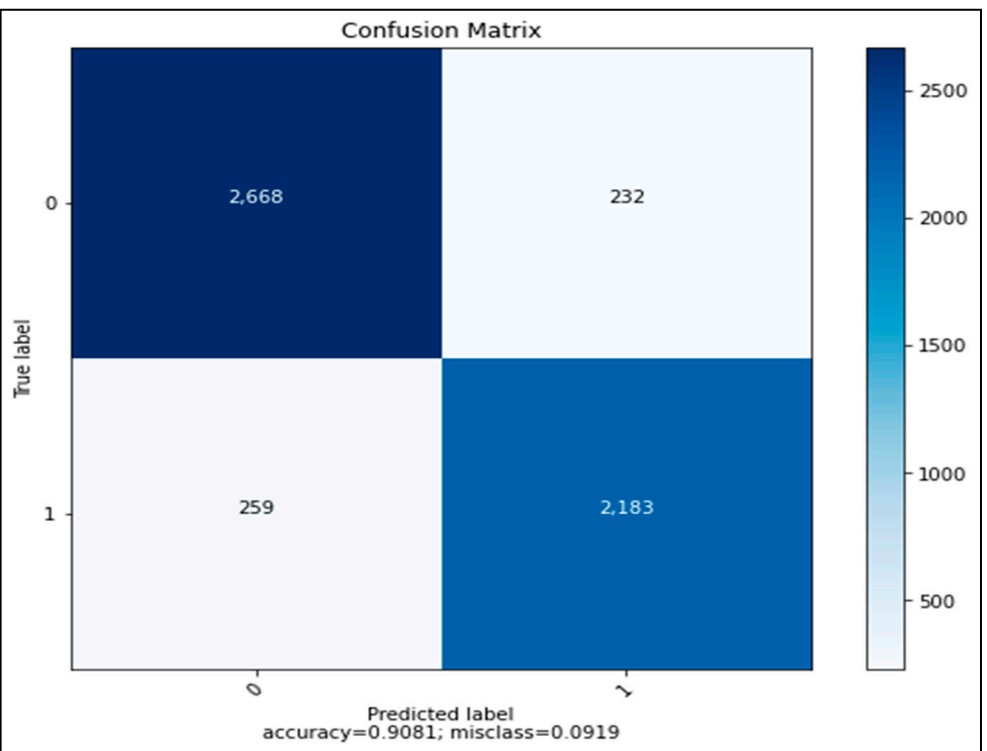

**Figure 16.** Confusion matrix results: Headlines dataset.

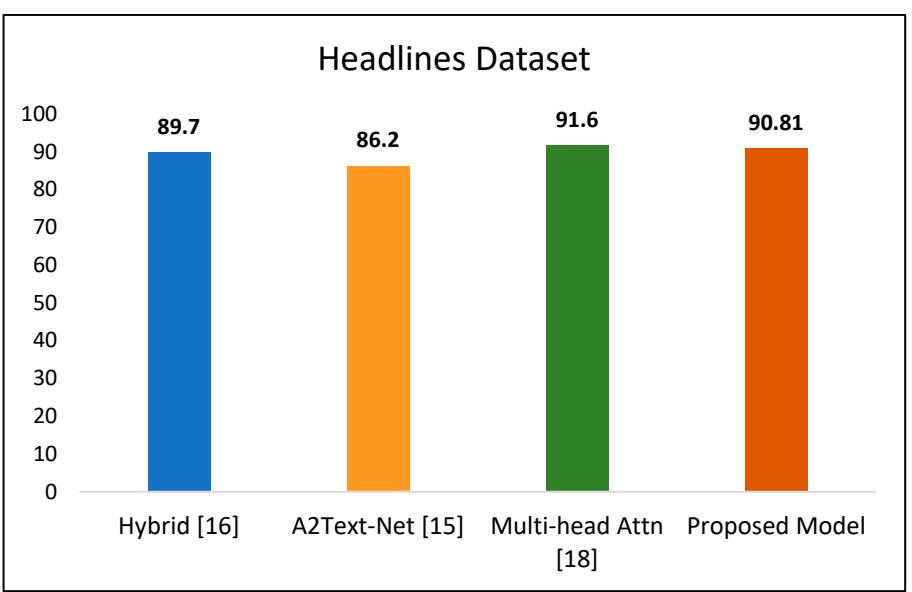

**Figure 17.** Accuracy evaluation: Headlines dataset [15,16,18].

To conclude, we state that having word-embedding methods such as GloVe and Word2Vec and sentence embedding methods such as BERT-base provides an overall generalization and robustness in the framework. When the data are short and crisp, such as in the

case of Twitter, Word2Vec and GloVe compensate for BERT-base, and in the case of multiple lines, in datasets such as Reddit or Headlines, BERT-base compensates for GloVe and Word2Vec. Moreover, the final classification employs a fuzzy logic layer in our proposed model. It balances out the drawbacks of individual techniques and assigns weights to them. Rules-based fuzzy logic works on the various combinations of the probabilities received from each method. There is less probability that any one of the techniques works better than others in all domains. Thus, the fuzzy logic layer handles each technique's strengths, and overall, the proposed framework performs better than previous frameworks. Research can be conducted using other techniques in the framework, such as gated ensemble or gated recurrent units; this will be taken up in further research by the authors of this study.

## 5. Conclusions

This study merges NLP-based word- and phrase-embedding methodologies to present a hybrid ensemble model for sarcasm detection. The model uses word and phrase embedding based on Word2Vec, GloVe, and BERT architecture. Each approach's embedding is learned through the dense layers, and classification probabilities are forecasted. The fuzzy logic evolutionary approach is given this projected probability. The assignment of weights to each possibility and classification of tweets or statements as sarcastic or not are performed by a fuzzy logic module based on fuzzy rules. This framework works well with a variety of content types and is ubiquitous. The model was evaluated using the publicly released datasets SARC, Headlines, and Twitter. The accuracy values were 85.38%, 90.81%, and 86.80% for the SARC, Headlines, and Twitter datasets, respectively. The higher accuracy is ascribed to the use of many approaches that cover different dataset types and the application of fuzzy rules that balance out the unique shortcomings of each methodology. The model may filter out sarcastic remarks from various businesses, social media mining specialists, and fact-checkers from a vast corpus. Sentiment analysis and opinion mining results can improve after the sarcastic remarks are made public. Additional language forms include parody, pun, and irony [46]. Tweets with similar sentence types can be observed for further research. Additionally, the dataset for headlines must be revised. Researchers might explore other evolutionary methods such as genetic algorithms for final analysis. Other techniques include gated recurrent units and gated ensembles, which can also be attempted with word/sentence embeddings.

**Author Contributions:** Conceptualization, D.K.S., B.S., S.A., N.P., A.A.A. and H.A.A.; Data curation, D.K.S., B.S., S.A., N.P., A.A.A. and H.A.A.; Formal analysis, D.K.S., B.S., S.A., N.P., A.A.A. and H.A.A.; Funding acquisition, A.A.A. and H.A.A.; Investigation, D.K.S., B.S., S.A., N.P., A.A.A. and H.A.A.; Methodology, D.K.S., B.S., S.A., N.P., A.A.A. and H.A.A.; Project administration, D.K.S., B.S., S.A., N.P., A.A.A. and H.A.A.; Resources, D.K.S., B.S., S.A., N.P., A.A.A. and H.A.A.; Software, D.K.S., B.S., S.A., N.P., A.A.A. and H.A.A.; Supervision, D.K.S., B.S., S.A., N.P., A.A.A. and H.A.A.; Validation, D.K.S., B.S., S.A., N.P., A.A.A. and H.A.A.; Visualization, D.K.S., B.S., S.A., N.P., A.A.A. and H.A.A.; Writing—original draft, D.K.S., B.S., S.A., N.P., A.A.A. and H.A.A.; Writing—review and editing, D.K.S., B.S., S.A., N.P., A.A.A. and H.A.A. All authors have read and agreed to the published version of the manuscript.

**Funding:** Princess Nourah bint Abdulrahman University Researchers Supporting Project number (PNURSP2023R 308), Princess Nourah bint Abdulrahman University, Riyadh, Saudi Arabia.

**Informed Consent Statement:** Informed consent was obtained from all subjects involved in the study.

**Data Availability Statement:** Not applicable.

**Acknowledgments:** Authors would like to give thanks for the support of Princess Nourah bint Abdulrahman University Researchers Supporting Project number (PNURSP2023R 308), Princess Nourah bint Abdulrahman University, Riyadh, Saudi Arabia.

**Conflicts of Interest:** The authors declare no conflict of interest.

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
