# Peer review of "Sarcasm Detection over Social Media Platforms Using Hybrid Ensemble Model with Fuzzy Logic"

_electronics, doi:10.3390/electronics12040937_

Round 1

Reviewer 1 Report

This manuscript presents a hybrid ensemble model for sarcasm detection. Generally, this manuscript is well prepared and in good formation. However, there are still some places that need further revisions or clarifications. The detailed comments are listed below for the authors' reference.

1. Some of the figures are not clear, especially the text in the pictures is too small and unclear.

2. Figure 7 is a table instead of a figure.

3. The formulae about Accuracy, Precision, Recall and F1-score metrics are missed.

4. Please add the existing latest models for comparison with the proposed model.

5. Experimental result should be analyzed and discussed, and the conclusions should be drawn.

6. The format of references needs to be unified.

7. In terms of computational burden, what are the advantages of the proposed model compared with the existing state-of-the-art model?

8. The fuzzy rule table is too simple and not specific.

Author Response

We thank the editor and the reviewers for their time and effort in evaluating the manuscript. We greatly appreciate all of the insightful comments and ideas which helped us improve the manuscript's quality. We have carefully considered the comments and made every effort to address each one.

This manuscript presents a hybrid ensemble model for sarcasm detection. Generally, this manuscript is well-prepared and in good formation. However, there are still some places that need further revisions or clarifications. The detailed comments are listed below for the authors' reference.

  1. Some of the figures are not clear, especially the text in the pictures is too small and unclear.

[Authors response] Authors acknowledge the comment. The paper has been updated. The following figures have been updated - Figures 3,4,9,15,16, and 17.

  1. Figure 7 is a table instead of a figure.

[Authors response] Authors acknowledge the comment. Actually, it's a figure of a table taken from a research paper [42]. The authors have updated the labelling of Figure 7 to mention it as a table.

  1. The formulae about Accuracy, Precision, Recall and F1-score metrics are missed.

[Authors response] Authors acknowledge the comment. The formulae for Accuracy, Precision, Recall and F1-score are updated in section 4.2 Metric after the confusion matrix Figure 11.

  1. Please add the existing latest models for comparison with the proposed model.

[Authors response] Authors acknowledge the comment. The authors did check other research papers mentioned in the paper, but most of them have not validated their frameworks on the same dataset as the authors. The current paper dataset is closer to social media-oriented datasets. All the previous research papers mentioned have been thoroughly checked, and if any of them have validated their framework on the same datasets, the results are already mentioned. Some papers with the same dataset, for example, Savini, E., Caragea, C. [27], have provided other metrics for comparison. Savini & Caragea have provided an F-1 score on the SARC dataset. Authors are comparing Accuracy as the primary metric with other research. But now authors have added the F1-score of Savini & Caragea in Table 3.

  1. Experimental result should be analyzed and discussed, and the conclusions should be drawn.

[Authors response] Authors acknowledge the comment. The section 4.3 Result Analysis has been updated to explain the analysis for each of the datasets and discuss why our proposed model metric is better than previous research. A concluding section at the end of section 4.3 is added to conclude the findings.

  1. The format of references needs to be unified.

[Authors response] Authors acknowledge the comment. The reference section had some issues. The reference section has been updated with correct numbering and format.

  1. In terms of computational burden, what are the advantages of the proposed model compared with the existing state-of-the-art model?

[Authors response] Authors acknowledge the comment. The proposed model is very light weight compared to other models as GloVe, Word2Vec and BERT-base are pre-trained models on NLP. Using transfer learning and updating them with sarcasm text learning is quick and less computation is required as they are pre-trained on large text corpus. The Fuzzy layer is also simple, and rule based. Thus, the entire architecture is light weight and very less computation intensive. Section 3.6 has been updated with the same.

  1. The fuzzy rule table is too simple and not specific.

[Authors response] Authors acknowledge the comment. The fuzzy logic used here is rule-based logic and hence looks simpler. But the combinations for high, low, and medium are based on the iterative experimental values. The authors have kept it rule based because of the requirement and also to keep the architecture lightweight and be flexible to be updated in case if used in  other scenarios like fake news detection, sentiment analysis etc.

Reviewer 2 Report

Automated processing of linguistic conversations is more than just Natural Language Processing (NLP) as under circumstances the obvious meaning is not what is meant. Sarcasm is one of a number of examples. The paper attempts to extend NLP for more universality. The presentation has no major issues though the illustration is too exuberant. It lists many facts on basic material though a bit more reasoning is highly appreciated.

The authors present a 3-level modular system. The pre-processing extracts the features of the sentences. Then the extracted variables are focused. Finally, the variables are interpreted to detect the hidden sarcasm. The authors review previous research in which the preprocessors find some but not all of the sarcasm. The ensemble technique is generally applied for the final selection.

The choice of the predecessor nets is explained by popularity. Actually, any choice would suffice if providing a sufficient set coverage. It is interesting to see whether there is a small number of optimal preprocessing nets, or any proper selection choice. Clearly the completeness (or at least sufficient overlapping) is mandatory or a mechanism is required to repair holes in the coverage.

The final selection is proposed with a Fuzzy System. The advantage is the linguistic control, but a neural selector is equally viable. There is a lot of theory of neural (Gated) Ensemble networks. For a more advanced set-up, this brings the potential of non-linear selection. The authors should spend some comments on such advancements. This makes the proposition a bit more complete.

The manuscript is extensively illustrated and supported with published papers from other authors. The amount of support material is too much for the basically short paper presented by the authors. It is advised to give a more personal knowledge understanding of the authors, or, at least, more chewed for easy digesting. Notably, the presented material lacks the factoring for educating the reader.

Author Response

We thank the editor and the reviewers for their time and effort in evaluating the manuscript. We greatly appreciate all of the insightful comments and ideas which helped us improve the manuscript's quality. We have carefully considered the comments and made every effort to address each one.

Automated processing of linguistic conversations is more than just Natural Language Processing (NLP) as under circumstances the obvious meaning is not what is meant. Sarcasm is one of a number of examples. The paper attempts to extend NLP for more universality. The presentation has no major issues though the illustration is too exuberant. It lists many facts on basic material though a bit more reasoning is highly appreciated.

[Authors response] Authors acknowledge the comment. Authors appreciate the findings. The linguistic term has been updated to figurative language in the abstract. The reasons for using these components are explained in section 1.2 major contribution and also in section 3.5 challenges in hybrid models.

The authors present a 3-level modular system. The pre-processing extracts the features of the sentences. Then the extracted variables are focused. Finally, the variables are interpreted to detect the hidden sarcasm. The authors review previous research in which the preprocessors find some but not all of the sarcasm. The ensemble technique is generally applied for the final selection.

[Authors response] Authors acknowledge the comment. Authors agree that some of the references provided are not directly linked to the topic, but they are included to provide a broad view of the problem statement and various techniques applied to resolve it. The proposed technique’s inherent novelty will be better understood if a little extra information is also provided to the readers for better understanding.

The choice of the predecessor nets is explained by popularity. Actually, any choice would suffice if providing a sufficient set coverage. It is interesting to see whether there is a small number of optimal preprocessing nets, or any proper selection choice. Clearly the completeness (or at least sufficient overlapping) is mandatory or a mechanism is required to repair holes in the coverage.

[Authors response] Authors acknowledge the comment. Coverage of various domains are taken care of by employing word and sentence embeddings techniques. The pre-trained GloVE is trained on Twitter, Wikipedia and Common Crawl massive web dataset. Word2Vec is pre-trained on Google News dataset (about 100 billion words). BERT-base is pre-trained on Wikipedia and Book Corpus. Thus, majorly all domains prominent on social media platforms are present in the pre-trained models.

The final selection is proposed with a Fuzzy System. The advantage is the linguistic control, but a neural selector is equally viable. There is a lot of theory of neural (Gated) Ensemble networks. For a more advanced set-up, this brings the potential of non-linear selection. The authors should spend some comments on such advancements. This makes the proposition a bit more complete.

[Authors response] Authors acknowledge the comment. Authors appreciate the feedback. Section 4.3 and conclusion has been updated to mention these techniques. These techniques can be used in future research over sarcasm detection.

The manuscript is extensively illustrated and supported with published papers from other authors. The amount of support material is too much for the basically short paper presented by the authors. It is advised to give a more personal knowledge understanding of the authors, or, at least, more chewed for easy digesting. Notably, the presented material lacks the factoring for educating the reader.

[Authors response] Authors acknowledge the comment. Section 1 Introduction and Section 3 explanation of GloVE, Word2Vec, BERT and Fuzzy logic has been updated in simple language for better understanding of the concepts. The experimental results Section 4.3 is better segmented and explanation of results on each dataset is updated. As mentioned in the first review comment, authors acknowledge the extra information from previous research but that it to provide a holistic view of the problem statement and other proposed solutions to the readers who are in the domain of sarcasm/fake new/sentiment detection.

Reviewer 3 Report

The explanation about classification method used need to be detailed. Since glove and word2vec for word representation, it is not clearly stated how the classification method work.

the fine-tuning configuration for bert and all method used must be stated in the paper

Author Response

The explanation about classification method used need to be detailed. Since glove and word2vec for word representation, it is not clearly stated how the classification method work.

[Authors response] Authors acknowledge the comment. Section 1 Introduction and Section 3 explanation of GloVE, Word2Vec, BERT and Fuzzy logic has been updated in simple language for better understanding of the concepts. The experimental results Section 4.3 is better segmented and explanation of results on each dataset is updated. Section 3.4 execution of the framework has been updated too.

the fine-tuning configuration for bert and all method used must be stated in the paper

[Authors response] Authors acknowledge the comment. Authors here have employed  GloVE, Word2Vec and BERT-base are pre-trained models. The pre-trained GloVE is trained on Twitter, Wikipedia and Common Crawl massive web dataset. Word2Vec is pre-trained on Google News dataset (about 100 billion words). BERT-base is pre-trained on Wikipedia and Book Corpus. So, not much is done at the model level. Section 4 first paragraph explains the hyperparameter values of the dense layers – “The model was created on a computer using the Google TensorFlow framework and the Keras library. The activation function "ReLU" and the optimizer "Adam" in the dense layers have a learning rate of 3e-4 each. Since this is a binary classification problem, binary-cross-entropy was employed as the loss function. For each component, these hyperparameters remained constant across all dense layers. The dataset was 20% for testing and 80% for training.”

Round 2

Reviewer 1 Report

All my questions have been well addressed by the author, and now I would like to recommend accepting the manuscript.